# A Nanofiber-Based Gas Diffusion Layer for Improved Performance in Air Cathode Microbial Fuel Cells

**DOI:** 10.3390/nano13202801

**Published:** 2023-10-21

**Authors:** Giulia Massaglia, Tommaso Serra, Fabrizio Candido Pirri, Marzia Quaglio

**Affiliations:** 1Department of Applied Science and Technology, Politecnico of Turin, Corso Duca degli Abruzzi 29, 10129 Torino, Italy; tommaso.serra@polito.it (T.S.); fabrizio.pirri@polito.it (F.C.P.); 2Center for Sustainable Future and Technologies, Italian Institute of Technology, Via Livorno 60, 10100 Torino, Italy

**Keywords:** electrospinning, fuel cell, laser-induced nanomaterials, microbial fuel cells, gas diffusion layer, triple phase boundary, oxygen reduction reaction

## Abstract

This work investigates a new nanostructured gas diffusion layer (nano-GDL) to improve the performance of air cathode single-chamber microbial fuel cells (a-SCMFCs). The new nano-GDLs improve the direct oxygen reduction reaction by exploiting the best qualities of nanofibers from electrospinning in terms of high surface-area-to-volume ratio, high porosity, and laser-based processing to promote adhesion. By electrospinning, nano-GDLs were fabricated directly by collecting two nanofiber mats on the same carbon-based electrode, acting as the substrate. Each layer was designed with a specific function: water-resistant, oxygen-permeable polyvinylidene-difluoride (PVDF) nanofibers served as a barrier to prevent water-based electrolyte leakage, while an inner layer of cellulose nanofibers was added to promote oxygen diffusion towards the catalytic sites. The maximum current density obtained for a-SCMFCs with the new nano-GDLs is 132.2 ± 10.8 mA m^−2^, and it doubles the current density obtained with standard PTFE-based GDL (58.5 ± 2.4 mA m^−2^) used as reference material. The energy recovery (EF) factor, i.e., the ratio of the power output to the inner volume of the device, was then used to evaluate the overall performance of a-SCMFCs. a-SCMFCs with nano-GDL provided an EF value of 60.83 mJ m^−3^, one order of magnitude higher than the value of 3.92 mJ m^−3^ obtained with standard GDL.

## 1. Introduction

The transition from carbon-based economies for sustainable human development is boosting renewable energy systems [1]. Alongside the more traditional technologies for the valorization of renewable energy, like wind, solar, hydro- and geo-thermal, and biomass, microbial fuel cells (MFCs) have gained great interest in recent years. MFCs are bio-electrochemical devices that combine a power production mechanism similar to that of traditional fuel cells with biotechnological processes like water treatment [2,3], bioremediation [4], and sensing [5,6,7], thanks to the presence of electroactive bacteria at their anodes [8,9,10,11,12]. These microorganisms can directly transduce the chemical energy entrapped in organic matter into electrical energy. Indeed, electroactive bacteria work as anodic biocatalysts in anaerobic conditions, oxidizing organic matter dissolved into the electrolyte and releasing the produced electrons to the anode. Electrons flow continuously to the cathode compartment, where their re-combination with terminal electron acceptors (TEAs) is ensured. MFCs in open-air configurations use oxygen as the TEA. The target cathodic reaction is the direct oxygen reduction reaction (ORR). As demonstrated by several works in the literature [13,14,15,16,17], ORR is the usual choice for the cathodic reaction, as oxygen maximizes power production while allowing the environmental application of MFCs. Catalysts are needed to promote the ORR according to the direct reduction of O_2_ in water and to optimize the reaction kinetics. If properly catalyzed, the secondary reaction pathway of the ORR, associated with the production of toxic hydrogen peroxide, can be completely avoided. The best-performing catalyst is platinum [13,14,15,16,17]. Pt is applied at the surface of the cathode in such a way as to ensure optimal contact with both the electrode itself and the electrolyte. The resulting interface is the most critical one for fuel cell technologies since it is associated with the triple-phase boundary, i.e., the zone where protons, electrons, and oxygen molecules must reach the catalyst sites to react [18,19,20,21]. In this view, the gas diffusion layer (GDL) is a key component of fuel cells to manage the diffusion of gaseous reactants, such as oxygen, to promote the removal of excess water in the proximity of the catalyst layer and, in MFCs, to minimize as much as possible the electrolyte leakage [22]. The ideal GDLs must satisfy several properties, such as high gas diffusion [23,24], good bending stiffness, continuous porosity, air permeability, water vapor diffusion, high surface-area-to-volume ratio to ensure water removal, and good electrical and electronic conductivity to ensure the proper electrons’ transfer and suitable mechanical stability [21]. Regarding air cathode single-chamber microbial fuel cells (a-SCMFCs), commonly used GDLs are characterized by a backbone made of a carbon-based material acting as the electrode, which is then covered by the catalyst layer on one side [25] and by a hydrophobic coating, typically based on polytetrafluoroethylene (PTFE) on the side. PTFE is strongly hydrophobic and permeable to oxygen. These properties can be very useful in designing GDLs for MFCs since this polymer can act as a barrier to avoid electrolyte leakage from the cell while allowing oxygen diffusion and contributing to avoiding excess water, thus preventing the cathode flooding [26,27,28,29,30]. Nevertheless, the PTFE layer needs careful design in order to avoid any negative influence on the final behavior of the cathode. Guerrini et al. [30] demonstrated that excess PTFE in GDLs for open-air cathode MFCs can make the electrode too hydrophobic, preventing water from reaching the catalytic sites and inhibiting the ORR reaction. Moreover, many works in the literature in the main fields of fuel cell technology focused their attention on the GDLs’ structure, which could be a bottleneck for improving the functionality of this layer [18,19,20,21,22,23,24,25,26,27,28,29,30]. Indeed, over the last decades, nanostructured materials have gained increasing interest in the design of GDLs to overcome the above-described limitations, ensuring optimized gas diffusion, optimal water management, and structural refinement [18,31,32,33].

The present work proposes the development of a novel nanostructured gas diffusion layer (nano-GDL) to improve the overall behavior of a-SCMFCs with the aim of obtaining the best compromise among hydrophobicity and surface wettability properties of GDL. To overcome all the limitations associated with the unprecise use of PTFE, which can induce an incomplete wetting of the cathode electrode, novel nano-GDLs were prepared by the electrospinning process directly collecting on the same carbon-based electrode two different nanofiber mats. The electrohydrodynamic process promotes optimal interaction among the different layers, avoiding the need for a binder. The first layer (inner layer) was made of cellulose nanofibers that play a crucial role in promoting oxygen diffusion into SCMFC [29,30]. With the aim of improving the adhesion of this first nanostructured layer to the carbon backbone, in the present work, we propose creating carbonized patterns in the cellulose nanofibers by direct laser writing. To this purpose, the carbonized patterns were designed, allowing the creation of graphene-like regions (i.e., laser-induced graphene) combined with untreated cellulose nanofibers, which played a key role in ensuring the necessary hydrophilicity to improve water retention in the proximity of the active catalytic sites, thus avoiding any decrease of proton conductivity of the electrolyte by dehydration [29,30]. The second layer (outward layer) was based on polyvinyl-fluoride (PVDF) nanofibers with the main purpose of preventing electrolyte leakage while allowing oxygen to flow freely combined with correct water removal from the cathode electrode. The design of the new nano-GDL allowed for the exploitation of all of the nanofibers’ intrinsic properties, such as high surface-area-to-volume ratio, high continuous porosity, and light weight, thus achieving a good oxygen diffusion in the proximity of the catalyst layer, ensuring optimal surface wettability, and thus favoring the direct ORR while preventing the water flooding in correspondence of catalyst layer. We proposed the ideal catalyst layer for ORR, based on platinum on carbon [34,35], directly deposited on the inner side of carbon paper, on which both of the two nanostructured layers were directly collected. To investigate the good performances of SCMFCs achieved when nano-GDLs were employed, cathode electrodes with commercial gas diffusion layers made of PTFE were used for comparison. We demonstrated the capability of a-SCMFCs with nano-GDLs to achieve a maximum current density equal to (132.2 ± 10.8) mA m^−2^, an order of magnitude higher than the one reached with commercial PTFE, equal to (58.5 ± 2.4) mA m^−2^. To confirm the extremely/excellent performances provided by nano-GDLs, we propose an analysis of obtained results in terms of energy recovery, as already reported in our previous work [36]. In line with the trend obtained by analyzing the current densities, it is possible to state that nano-GDLs ensured the achievement of an energy recovery of 60.83 mJ m^−3^, one order of magnitude higher than the value obtained by commercial PTFE (3.92 mJ m^−3^). All these latter results open the doors to the design of the whole nanostructured cathode electrode in SCMFCs. To achieve these goals, nitrogen-doped carbon nanofibers (N-CNFs) can be proposed as conductive carbon backbone, simultaneously able to exploit good electrocatalytic properties for ORR, as deeply defined in our previous work [11].

## 2. Materials and Methods

### 2.1. Gas Diffusion Layer Package and Nanofibers Synthesis

Nanostructured GDLs were obtained by deposition of nanofiber mats directly onto carbon-based materials (carbon paper was purchased from Fuel Cell Earth). To this purpose, electrospinning process (NANON 01A electrospinning apparatus MECC, LTD.) was involved in designing nano-GDLs, guaranteeing the ability to connect two different nanofibers’ layers without the use of a binder, suitable to connect all developed layers to the carbon backbone. In this configuration, carbon paper pieces, cut in a square shape of (3 × 3) cm^2^, were used as counter electrodes during the electrospinning process, thus leading to obtaining cathode electrodes with the desirable geometric area for application in microbial fuel cells. Nano-GDLs were composed of two different layers. The first layer was based on cellulose nanofibers obtained by implementing a hydrolyzation process of cellulose acetate nanofibers [37,38]. Indeed, cellulose acetate nanofibers were immersed for 1 h in an ethanol-based solution containing 0.05 M di NaOH. To guarantee a successful carbonization process in ambient atmosphere, no subsequent washing is needed. In this way, it has been possible to perform the laser writing step under ambient conditions, directly obtaining the LIG patterns from the cellulose nanofibers. This experimental approach improves the adhesion of all nanostructured GDLs to the carbon backbone. CO_2_ laser writing (pulsed CO_2_ laser source, implemented by commercial Laser Scriber by Microla Optoelectronics S.r.l) was employed to conduct the carbonization of cellulose nanofibers, as deeply investigated in the literature [37,38]. In particular, the great advantage of CO_2_ laser writing was identified by the capability to create carbonized patterns on nanofiber mats, following different and desirable paths. We proposed the possibility of performing the LIG step under ambient conditions, writing onto cellulose nanofibers carbonized patterns, which led to an improvement in the adhesion of all nanostructured GDLs to the carbon backbone. The second layer was made of polyvinyl fluoride, PVDF-nanofibers, achieved by starting a polymeric solution of 2 g of PVDF (Mw = 150 kDa, Sigma-Aldrich, Steinheim, Germany) dissolved in a mixture (1:1 *v*/*v*) of N-N Dimethylformamide (N-N DMF) and acetone. The deposition of PVDF nanofibers is achieved to ensure the correct balance between hydrophilic properties and hydrophobic features, preventing possible leakage of electrolyte solution.

For both nanofibers’ mats, the electrospinning parameters were an applied voltage of 26 kV, a working distance of 15 cm, and a flow rate of 0.5 mL h^−1^. 

### 2.2. SCMFCs Architecture and Operation

The experiment was carried out using squared-shaped open-air cathode SCMFCs that we discussed in previous articles [10,11]. A 3D printer (OBJET 30) was used to fabricate the membrane-less cells, characterized by a single reaction chamber shared between anode and cathode. The two electrodes were kept at a fixed distance thanks to an intermediate septum, ensuring an inner volume of 12.5 mL. Both the electrodes have a squared shape, with a geometric surface area close to 5.76 cm^2^. They were both made of carbon paper (CP, from, AvCarb, Lowell, MA, USA). To ensure maximum anodic stability, all the anodes of the proposed experiment have been obtained from a-SMFCs already running in our labs [36], characterized by anodic biofilms made of mixed consortia. For what concerns the cathode electrodes, on the inner side of cathode electrode, standard platinum catalyst layer made of 0.5 mg cm^−2^ of Pt/C (by Sigma-Aldrich, St. Louis, MO, USA) and 5 wt% of Nafion (Sigma-Aldrich) was applied [34]. On the outer side of cathode electrode, with the main purpose of improving the triple phase boundary to ensure the direct ORR and consequently increasing the overall devices’ performance, three different GDLs were proposed and compared: (i) The first one was made of a nanostructured GDL and was named nano-GDL. It consisted of 2 layers of nanofiber mats, with the first one being made of hydrolyzed acetate nanofibers, on which a second layer of PVDF-nanofibers was deposited. (ii) The second one, defined as nano-LIG GDL, was based on two different layers of nanofibers: a layer of cellulose nanofibers patterned by LIG and a PVDF-nanofibers layer. (iii) The third material was composed of a polytetrafluoroethylene (PTFE) layer, and it is referred to as commercial PTFE. A water-based solution containing 12 mM of sodium acetate was used as the electrolyte, with sodium acetate serving as the carbon energy source. Ammonium chloride (5.8 mM) and phosphate-buffered saline solution (PBS) [10,11,12] were added to promote the metabolic activity of microorganisms. Titanium wires were employed for electrical contact, and a multichannel data acquisition unit (Agilent 34972A) was used to monitor the output voltage from the cells. Titanium wires were threaded through the polymeric frame so that once the frame is correctly positioned inside the cathode compartment and exploiting the mechanical pressure obtained when closing the cell, an optimum electrical contact is ensured. An external load of 1 kΩ was applied to each cell. Indirectly, through the first Ohmic Law, current values were defined. Moreover, a current density trend was obtained by dividing each current value by the geometric area of the electrode, which is equal to 5.76 cm^2^. All the devices were operated in fed-batch mode, thus ensuring the electrolyte replacement when the output voltage dropped to 0 V. All the experiments were conducted in duplicate.

### 2.3. Characterizations and Measurements

Field Emission Scanning Electron Microscopy (FESEM, Supra operating from 5 kV to 10 kV) is used to evaluate the morphological properties of nanostructured gas diffusion layer (GDL). The analysis of FESEM images allowed for defining the preservation of nanostructures after the LIG process, implementing to improve the adhesion of both nanofibers’ layers, components of the new gas diffusion layer GDL. Moreover, to confirm the effective role of high continuous porosity in the diffusion of oxygen species inside the devices and on the balance between water retention and removal, FESEM images were analyzed with imaging software (ImageJ, Version 1.53t). 

Final porosity of both samples, nano-LIG GDL and commercial PTFE, were indirectly determined by dividing the volume *V* occupied by nanofiber or PTFE layer, respectively, by total volume (*V_tot_*).
(1)$$\phi =\frac{V}{V_{tot}}$$

Furthermore, to confirm the effectiveness of imageJ analysis performed on FESEM images, Brunauer–Emmett–Teller (BET, ASAP 2020 Plus model, Micrometrics, Cumming, GA, USA) measurements were implemented to define the specific surface area of nano-LIG GDLs, thus leading to the evaluation of the porosity distribution of the nanostructured layers.

To evaluate how nano-LIG GDL can affect SCMFCs performance, improving it with respect to the performance achieved with nano-GLD and commercial PTFE, Linear Sweep Voltammetry (LSV) characterizations were defined by using Palmsens potentiostat (Palmsens4, Houten, The Netherlands). LSV characterizations were performed at the end of experimental study, with the voltage ranging from open circuit to a short circuit at a rate of 0.1 mV s^−1^. Electrochemical impedance spectroscopy (EIS), using a Palmsens potentiostat, was provided to define electrocatalytic features of cathodes. EIS analysis was performed by applying an external fixed resistor of 100 Ω [12], using a sinusoidal signal with an amplitude of 25 mV and frequency spanning between 150 kHz and 200 mHz.

The equivalent circuit of Figure 1 was used to fit the experimental data from EIS in such a way to quantitatively evaluate the following electrical parameters: (i) R1, i.e., the series resistance, accounts for electrolyte and wiring resistances. (ii) R2 and R3 are associated with the resistances to charge transport inside the electrode and to the charge transfer at the electrode/electrolyte interface, respectively. Due to porous nature of cathode electrodes, constant phase elements Q1 and Q2 are used to model the corresponding double-layer capacitances [12]. And finally, (iii) Warburg element was included to model low-frequency features, commonly corresponding to the species diffusion.

## 3. Results and Discussion

### 3.1. Morphological Properties of Nano-GDL and Commercial-PTFE

Aiming to perform the LIG step in ambient conditions to facilitate the fabrication process to scale it up, the main results were achieved by the definition of proper conditions for graphitization, avoiding the burning of the material. Indeed, to ensure the carbonization of cellulose nanofibers, high temperatures must be reached, at which these synthetic polymers basically burn in the presence of oxygen. All these limitations demand the development of a polymeric nanofibers treatment capable of allowing and guaranteeing the carbonization of the nanofibers themselves under ambient conditions, without the implementation of technical gases, such as argon or nitrogen, typically employed during the standard carbonization process [36]. The obtained results highlighted how the deacetylation step, usually involved in transforming the cellulose acetate nanofibers into cellulose nanofibers, if not followed by subsequent washing in deionized water, resulted in being pivotal to ensuring the carbonization of nanofibers under ambient conditions. It was possible to demonstrate that the presence of NaOH salts in cellulose nanofibers had a key role in promoting the carbonization of cellulose nanofibers. As shown in Figure 1a, the presence of NaOH salt decorating cellulose nanofiber mats was confirmed by FESEM images, whereas on the contrary, NaOH completely disappeared after the implementation of the LIG process conducted in ambient conditions, as represented in Figure 1b. Moreover, Figure 1b demonstrates the preservation of nanostructures after the LIG process is applied to create carbonized patterns on nanofiber mats.

This result can be explained by considering the hypothesis that the NaOH salts were able to sublime during the CO_2_ laser writing process, ensuring the formation of an oxygen-free atmosphere during the process itself, thus leading to the complete transformation of cellulose nanofibers into carbon-based nanofibers. With the main purpose of confirming the capability of CO_2_ laser writing to transform into laser-induced graphene (LIG) nanofibers, the initial cellulose nanofibers, Raman characterization was employed. As reported in Figure 2, it is possible to detect an achieved graphitization of material as evidenced by the presence of the G peak at about 1580 cm^−1^. The graphitization is certainly partial, and the presence of a pronounced ‘neck’ between the G peak and the D peak (at about 1350 cm^−1^) indicates the presence of groups of various kinds bound to the graphitic lattice. These groups can be defined as oxygen-containing functional groups due to the partial graphitization of the starting cellulose nanofibers [37]. Regarding the 2D band peak, this peak is not detectable due to the low degree of graphitization, thus excluding this part of the spectrum because it does not provide any further information.

Moreover, the morphological properties of the whole nano-LIG GDLs, obtained after the deposition of PVDF-nanofiber layers on the cellulose nanofibers, previously carbonized through the CO_2_ laser writing process, were investigated and reported in Figure 3a, thus highlighting the pore distribution in these samples. Indeed, nano-LIG GDLs are characterized by pores with dimensions in the range of a few micrometers, thus leading to the exhibition of a higher surface-area-to-volume ratio than the one obtained with commercial gas diffusion layers made of PTFE (see Figure 3b). The ImageJ software allowed for the indirect evaluation of the porosity distribution into nano-LIG GDL, comparing it with the one achieved with commercial PTFE. To this purpose, the porosity distribution of all samples was determined by applying Equation (1). Figure 3c allowed for the confirmation that the porosity distribution of nano-LIG GDL, close to 77%, results in being higher than the one when commercial PTFE was applied (equal to 25%). High porosity distribution values are in line with the results achieved by determining the specific surface area of whole nano-LIG GDLs by implementing Brunauer–Emmett–Teller (BET) measurements. Nano-LIG GDLs showed a specific surface area of 18.94 ± 0.05 m^2^/g. Moreover, from BET measurements, it is possible to define the distribution of the pores inside the whole nanostructured layers. Nano-LIG GDLs showed pore dimensions in the range from 2 nm to 10 nm, thus leading to the definition that nano-LIG GDL is a micro and mesoporous material. High continuous porosity shown in Figure 3c, greater than the one achieved by commercial PTFE, can positively affect the diffusion of oxygen from outside towards the catalytic active sites, thus leading to an improvement in the triple phase boundary. These intrinsic properties of nanofibers, with particular attention paid to their high continuous porosity, can play a pivotal role in the enhancement of oxygen diffusion in the proximity of the triple contact zone, thus ensuring better oxygen transport in the proximity of catalytic active sites.

### 3.2. SCMFCs Performance

As previously described, anode electrodes on which a microorganism’s proliferation occurred were obtained by the preceding experiments [36]. For all cathode electrodes, platinum is involved as a catalyst, while nano-LIG GDL, nano-GDL, and commercial PTFE were compared to evaluate how all the properties of both nano-GDLs can affect the overall performance of SCMFCs. Figure 4 represents the current density trends over time. Nano-LIG GDL reached a maximum current density equal to (151.2 ± 15.5) mA m^−2^, which is double the one reached with nano-GDL, confirming how the presence of conductive patterns played a pivotal role in ensuring the design of a better-performing gas diffusion layer. Moreover, Figure 4 also highlighted that the nanostructured arrangement of gas diffusion layers allowed for achieving improved overall SCMFC performance compared with commercial PTFE (maximum current density close to (58.5 ± 2.4) mA m^−2^) used as reference material. Since the anode electrodes are formally identical for all devices, it is possible to attribute the different power production of SCMFCs directly to the development of diverse nano-GDLs on cathode electrodes. 

This latter result confirms the key role of nano-LIG GDL, with a great interest in the presence of conductive patterns created through the CO_2_ laser writing, able to further improve overall SCMFCs’ performance. Indeed, the presence of conductive patterns could improve the contact between the whole nanostructured GDL and the carbon backbone, allowing for the enhancement of the triple contact zone, oxygen diffusion in the proximity of catalytic active sites, and, at the same time, thanks to the presence of hydrophilic cellulose-NFs layers, the excess water removal from catalytic sites. The combination of these features affects the electrocatalytic efficiency of the electrode toward direct ORR, thus reflecting the augmented overall SCMFCs’ performance. In line with the obtained results, we performed LSV and electrochemical characterizations on SCMFCs that showed nano-GDL, nano-LIG GDL, and commercial PTFE. 

Both SCMFC devices reach a similar open circuit voltage (OCV) close to 0.4 V, while SCMFCs with nano-LIG GDL achieved a higher short circuit current (close to 262 ± 5 mA m^−2^) than the one obtained with commercial PTFE (176.4 ± 3.2 mA m^−2^). Since nano-LIG GDL can affect and favor the direct ORR, ensuring the best oxygen diffusion into SCMFCs, the variation of total cathodic resistance over time was investigated through EIS. Typical Nyquist plots are represented in Figure 5b, comparing nano-LIG GDL and the commercial one. The curves obtained by the fitting procedure are overlaid on the experimental data (see Figure 5b). Table 1, on the contrary, summarizes all resistance values.

As highlighted in Figure 5b, all SCMFCs show a similar value of series resistance R_1_ independently of cathode electrodes. This is to be expected since electrolytes, wires, and electrical connections are identical for all SCMFCs. Moreover, the lower the charge transfer at the electrode/electrolyte interface R_3_, the higher the capacity of the cathode electrode to ensure a faster electron flow. This result demonstrates the effectiveness of nano-LIG GDL to ensure an enhancement of oxygen diffusion, consequently improving the occurrence of direct ORR and the overall performance of SCMFCs. A similar trend is observed for the transport resistance R_2_, which is visible in the high-frequency smaller arc sketched in Figure 5b. 

A lower R_2_ defines an increase in electrode transport properties, and since all other aspects of cathode electrodes are the same, it is possible to confirm how nano-GDL results are more efficient in carrying out ORR. 

Moreover, in line with all obtained results, the analysis performed in terms of energy recovery parameter, defined as the ratio of generated power integral and the internal volume of devices, allows for evaluating the overall SCMFC performance. SCMFCs with a nano-LIG GDL showed an energy recovery equal to 60.83 mJ m^−3^, which was one order of magnitude higher than the one obtained with commercial PTFE, close to 3.92 mJ m^−3^.

## 4. Conclusions

In the present work, nano-LIG GDL was designed as a new gas diffusion layer to improve the oxygen diffusion inside SCMFCs, exploiting the intrinsic properties of nanofibers, such as high porosity, high surface-area-to-volume ratio, and light weight. Moreover, through the electrospinning process, nanofiber mats were directly collected on carbon-based materials used as cathode electrodes without the necessity of a binder to bond GDL with the carbon backbone. Morphological analysis provided evidence of a higher porosity obtained with nano-GDL than that reached using commercial PTFE.

All obtained results demonstrate that nano-LIG GDLs are effective in improving a-SCMFCs performance, showing a maximum current density value that doubles the one obtained using standard PTFE. Since the changes introduced among the cathodes used in this work are referred to as the new nano-LIG GDLs, it is possible to confirm that the observed improvement is related to the use of nanostructured materials for GDL. All the obtained results demonstrate the effectiveness of nano-LIG GDL to ensure an enhancement of oxygen diffusion, consequently enhancing the direct ORR and, thus, the final performance of a-SCMFCs.

## Data Availability

Not applicable.

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
