# Peer review of "A Nanofiber-Based Gas Diffusion Layer for Improved Performance in Air Cathode Microbial Fuel Cells"

_nanomaterials, 2023, doi:10.3390/nano13202801_

Round 1

Reviewer 1 Report

The relevance of the research is primarily related to the development and improvement of new energy sources, increasing the efficiency of MFCs.

The article has a generally accepted structure and is executed according to all requirements of the journal.

After reading the manuscript, a few questions arose:

1. In the Raman spectrum of the obtained composite material, there is a pronounced shoulder in the region of 1400-1450 cm-1. This may be due to the formation of graphene oxide. Did the authors confirm the presence or absence of graphene oxide phase? In addition, it is worth citing the spectrum in the region above 2000 cm-1, important for the characterization of the 2D band.

2. The porosity of the composite was determined indirectly by micrographs. However, this method allows estimation of the surface porosity. Did the authors use methods to determine membrane porosity, such as nitrogen adsorption/desorption. Is total porosity an important characteristic for MCFs?

After correcting the deficiencies, the article could be published.

Author Response

Dear Reviewer, 

we would like to thank you  for the thorough evaluation and interest in the paper. Given below are the answers to your suggestions.

All the changes made in the revised version of the paper are highlighted in red in the response and in the manuscript.

We hope that the paper could now be suitable for publication.

Sincerely,

Giulia Massaglia

The relevance of the research is primarily related to the development and improvement of new energy sources, increasing the efficiency of MFCs.

The article has a generally accepted structure and is executed according to all requirements of the journal.

After reading the manuscript, a few questions arose:

  1. In the Raman spectrum of the obtained composite material, there is a pronounced shoulder in the region of 1400-1450 cm-1. This may be due to the formation of graphene oxide. Did the authors confirm the presence or absence of graphene oxide phase? In addition, it is worth citing the spectrum in the region above 2000 cm-1, important for the characterization of the 2D band.

We thank the Reviewer for these considerations that allowed us to better explain obtained results related to Raman spectrum.  Raman spectrum was implemented to investigate the creation of carbonized patterns, trough the employment of LIG process, directly applied onto cellulose nanofibers. In particular, by analysing Raman spectrum, it is possible to detect an achieved graphitisation of material as evidenced by the presence of the G peak at about 1580 cm-1. The graphitisation is certainly partial, and the presence of a pronounced 'neck' between the G peak and the D peak (at about 1350 cm-1) indicates the presence of groups of various kinds bound to the graphitic lattice. These groups can be defined as oxygen-containing functional groups associated to the partial graphitization of the starting cellulose nanofibers, as suggested by Ref 37 of the manuscript. Whereas it would be inaccurate to attribute the presence of graphene oxide within the graphitic lattice to this peak. For what concerned the 2D band peak, this peak is not detectable due to the low degree of graphitisation, leading thus to exclude this part of spectrum, because it does not provide any further information.

In the Results and Discussion section, we modify the main manuscript accordingly to comprehensively describe the obtained Raman spectrum.

  1. The porosity of the composite was determined indirectly by micrographs. However, this method allows estimation of the surface porosity. Did the authors use methods to determine membrane porosity, such as nitrogen adsorption/desorption. Is total porosity an important characteristic for MCFs?

Following the consideration of the Reviewer, we add BET measurements performed on nano-LIG GDLs, estimating not only the specific surface area of nano-LIG GDLs, but, at the same time, the porosity distribution and sizes obtained for whole nanostructured layers. The BET measurements, in our opinion, complement and confirm the considerations that can be made when considering porosity indirectly calculated through the analysis of morphological properties.

Starting from this consideration, we can better explain how the data on the porosity were obtained. The morphological properties, represented in Figure 3a), characterize nanostructured-LIG GDLs, namely the GDLs made of all nanostructured layers, based on the PVDF-based nanofibers collected on the cellulose nanofibers, previously carbonized through CO2 laser writing process. High continuous porosity showed in Figure 3c), greater than the one achieved by commercial-PTFE, can positively affect the diffusion of oxygen from outside towards the catalytic active sites, leading thus to improve the triple phase boundary.

To this purpose, the ImageJ software allow us defining the porosity distribution into Nano-LIG GDL, resulting to be higher than the one ensured by commercial carbon paper that showed a PTFE layer to promote oxygen diffusion. High porosity distribution values are in line with the results achieved by determining specific surface area of whole nano-LIG GDLs by implementing Brunauer–Emmett–Teller (BET) measurements. Nano-LIG GDLs showed a specific surface area of 18.94±0.05 m2/g.  Moreover, from BET measurements, it is possible to define the pores distribution inside the whole nanostructured layers. Nano-LIG GDLs showed pores dimension in the range from 2nm to 10 nm, leading thus to define that Nano-LIG GDLs is a micro and mesoporous material.

The main manuscript was modified accordingly.

After correcting the deficiencies, the article could be published.

Reviewer 2 Report

The article corresponds to the topic of the journal. The work proposes methods for applying a two-layer coating of nanofibers to a carbon carrier; such composite materials are of interest for use in fuel cells. The paper describes a number of interesting data on the technology of forming gas-permeable coatings using two types of nanofibers with different functional properties.

Unfortunately, the description of the method for obtaining the material presented in the article is too general, contains many references to the literature and does not make it possible to understand the real conditions of syntheses, as well as the structure and properties of the formed composite:

- what is the geometry of the substrates used in the work, in what atmosphere was the procedure of graphitization of coatings under the influence of laser radiation carried out, what is the morphology of the layer formed by this method on the surface of the carbon carrier, etc.

- judging by the information given in the article, the applied coating is quite heterogeneous. To which part of it do the spectra shown in Figure 2 belong? For comparison, it would make sense to present here the spectra of the sample before the graphitization procedure.

- it should be indicated how the data on the porosity of the samples shown in Figure 3c) were obtained and how this characteristic changes after carbonization of the samples.

- it is necessary to provide data on the methods of applying contacts used in the work, their geometry and the conditions for conducting electrical measurements, data on which are shown in Figure 4.

In general, the article in its presented form cannot be published and requires significant revision.

Author Response

Dear Reviewer, 

we would like to thank you  for the thorough evaluation and interest in the paper. Given below are the answers to your suggestions.

All the changes made in the revised version of the paper are highlighted in red in the response and in the manuscript.

We hope that the paper could now be suitable for publication.

Sincerely,

Giulia Massaglia

The article corresponds to the topic of the journal. The work proposes methods for applying a two-layer coating of nanofibers to a carbon carrier; such composite materials are of interest for use in fuel cells. The paper describes a number of interesting data on the technology of forming gas-permeable coatings using two types of nanofibers with different functional properties.

Unfortunately, the description of the method for obtaining the material presented in the article is too general, contains many references to the literature and does not make it possible to understand the real conditions of syntheses, as well as the structure and properties of the formed composite:

- what is the geometry of the substrates used in the work, in what atmosphere was the procedure of graphitization of coatings under the influence of laser radiation carried out, what is the morphology of the layer formed by this method on the surface of the carbon carrier, etc.

We would like to thank the Reviewer for all these considerations, allowing us to give all required information to improve the description of experimental adopted methods in the present work. 

Nanostructured GDLs, with and without the implementation of LIG process, are obtained by directly depositing nanofibers mats onto carbon-based materials, without the employment of a binder to connect all nanostructured layers onto the substrate.  This is an intrinsic advantage of electrospinning process, that allows implementing as counter electrode directly the useful final substrate. In our experimental work, the substrate is a carbon-based material, which is a carbon paper (purchased from Fuel Store, USA) cut in pieces with a squared shape with the dimension of (3x3) cm2. This dimension allows us to obtain a final cathode electrode with the geometric area of 5.76 cm2, which is the typical one of MFCs employed in this work.

The main manuscript, in the paragraph 2.1 Gas Diffusion Layer package and nanofibers synthesis of Materials and Methods, was modified as followed:

“In this configuration, carbon paper pieces, cut in a square shape of (3x3) cm2, were used as counter electrodes during the electrospinning process, leading thus to obtain cathode electrodes with the desirable geometric area for application in Microbial Fuel Cells.” The carbonization process, obtained by implementing CO2 Laser Writing, is carried out at ambient atmosphere, without the necessity to use technical gases, such as argon or nitrogen, typically employed during standard carbonization process, named pyrolysis treatment.

To better explain this concept, in the paragraph 2.1 Gas Diffusion Layer package and nanofibers synthesis of Materials and Methods, the following sentence are added:

“To guarantee a successful carbonization process in ambient atmosphere, no subsequent washing is needed. In this way, it has been possible to perform the laser writing step under ambient conditions, directly obtaining the LIG patterns from the cellulose nano-fibers. This experimental approach improves the adhesion of all nanostructured GDLs to the carbon backbone.”.

Finally, as stated in the Results and Discussion section, CO2 laser writing guarantee the carbonization of cellulose nanofibers, leading thus to demonstrate the key role of deacetylation step that results to be essential to ensure the carbonization of nanofibers. As reported in Figure 1b, it is possible to demonstrate the preservation of nanostructures also after the LIG process, applied to create carbonized patterns onto nanofibers mats.

- judging by the information given in the article, the applied coating is quite heterogeneous. To which part of it do the spectra shown in Figure 2 belong? For comparison, it would make sense to present here the spectra of the sample before the graphitization procedure.

We thank the Reviewer for these considerations that allowed us to better explain obtained results related to Raman spectrum.  Raman spectrum was implemented to investigate the creation of carbonized patterns, trough the employment of LIG process, directly applied onto cellulose nanofibers. In particular, by analysing Raman spectrum, it is possible to detect an achieved graphitisation of material as evidenced by the presence of the G peak at about 1580 cm-1. The graphitisation is certainly partial, and the presence of a pronounced 'neck' between the G peak and the D peak (at about 1350 cm-1) indicates the presence of groups of various kinds bound to the graphitic lattice. These groups can be defined as oxygen-containing functional groups. Whereas it would be inaccurate to attribute the presence of graphene oxide within the graphitic lattice to this peak. For what concerned the 2D band peak, this peak is not detectable due to the low degree of graphitisation, leading thus to exclude this part of spectrum, because it does not provide any further information.

In this work Raman spectroscopy is implemented to investigate the graphitic degree of carbon patterns obtained by processing cellulose nanofibers with CO2 laser writing 3under ambient condition, following the process proposed by Lee et al (see Ref 37 of the manuscript). Since the experimental method that we used in this work is analogous to the one optimized in that article, we directly compared Raman spectra for the newly developed nanostructured-LIG GDLs to the spectra reported in the Ref 37.  For this reason, the Raman spectra before laser-induced graphitization are not reported.

In the Results and Discussion section, we modify the main manuscript accordingly to comprehensively describe the obtained Raman spectrum.

- it should be indicated how the data on the porosity of the samples shown in Figure 3c) were obtained and how this characteristic changes after carbonization of the samples.

Starting from this consideration, we can better explain how the data on the porosity were obtained. The morphological properties, represented in Figure 3a), characterize nanostructured-LIG GDLs, namely the GDLs made of all nanostructured layers, based on the PVDF-based nanofibers collected on the cellulose nanofibers, previously carbonized through CO2 laser writing process. High continuous porosity showed in Figure 3c), greater than the one achieved by commercial-PTFE, can positively affect the diffusion of oxygen from outside towards the catalytic active sites, leading thus to improve the triple phase boundary.

To this purpose, the ImageJ software allow us defining the porosity distribution into Nano-LIG GDL, resulting to be higher than the one ensured by commercial carbon paper that showed a PTFE layer to promote oxygen diffusion. High porosity distribution values are in line with the results achieved by determining specific surface area of whole nano-LIG GDLs by implementing Brunauer–Emmett–Teller (BET) measurements. Nano-LIG GDLs showed a specific surface area of 18.94±0.05 m2/g.  Moreover, from BET measurements, it is possible to define the pores distribution inside the whole nanostructured layers. Nano-LIG GDLs showed pores dimension in the range from 2nm to 10 nm, leading thus to define that Nano-LIG GDLs is a micro and mesoporous material.

- it is necessary to provide data on the methods of applying contacts used in the work, their geometry and the conditions for conducting electrical measurements, data on which are shown in Figure 4.

We thank the Reviewer for this consideration that allowed us defining better that Figure 4 reports current density trend vs time, obtained by applied all proposed materials into the cathode compartment of Microbial Fuel Cells. During this experiment, we apply nanostructured GDL, on the outer side of cathode electrodes, and it is directly compared with the commercial PTFE, usually applied at the outer side of carbon paper. In this work, carbon paper was used as carbon backbone for both investigated GDLs, nano-GLD, nano-LIG GDL and commercial-PTFE. Whereas, on the inner side of carbon paper a catalyst layer, made of 0.5 mg cm-2 of Pt/C (by Sigma Aldrich) and 5wt% of Nafion (Sigma Aldrich), is applied. To monitor the electrical output achieved by Microbial Fuel Cells, titanium wires were threaded as electrical contacts. As reported in the  Figure, reported in the attached files, it is possible to appreciate how the titanium wires are threaded through the polymeric frame, so that once the frame is correctly positioned inside the cathode compartment and exploiting the mechanical pressure obtained when closing the cell, an optimum electrical contact is ensured. An external load of 1 kΩ is applied and a multichannel acquisition system was used to monitor and register the voltage output. Indirectly, through the first Ohmic Law, current values are defined. Moreover, a current density trend is obtained by dividing each current value for the geometric area of electrode, equal to 5.76 cm2. To this purpose the overall device performances are evaluated for the entire period time of experiment, close to 1 month. Our bio-electrochemical devices work under ambient conditions and by implementing a fed-batch mode.

In the Materials and Methods section, we modify the main manuscript better explaining the use of Ti wires to fabricate the contacts.

Round 2

Reviewer 2 Report

The necessary clarifications and additions have been made to the article. It may be published in its presented form.